

# ArduiTaM: accurate and inexpensive NMR auto tune and match system

Mazin Jouda[*1], Saraí M. Torres Delgado[*1], Mehrdad Alinaghian Jouzdani[1], Dario Mager[1], and Jan G. Korvink[1]

[1]Karlsruhe Institute of Technology (KIT), Institute of Microstructure Technology, Karlsruhe 76131, Germany

**Correspondence:** Jan G. Korvink (korvink@kit.edu)

**Abstract.** We introduce a low complexity, low cost, yet sufficiently accurate automatic tune and match system for NMR and MRI applications. The ArduiTaM builds upon an Arduino Uno embedded system that drives a commercial frequency synthesizer chip to perform a frequency sweep around the Larmor frequency. The generated low power signal is fed to the NMR coil, after which the reflected waves are detected using a directional coupler, and amplified. The signal shape is then
extracted by means of an envelope detector and passed on to the Arduino, which performs a dip search while continuously generating actuator control patterns to adjust the tune and match capacitors. The process stops once the signal dip reaches the Larmor frequency. The ArduiTaM works readily with any spectrometer frequency in the range from $1\,\mathrm{T}$ to $23\,\mathrm{T}$. The speed of ArduiTaM is mainly limited by the clock of the Arduino, and the capacitor actuation mechanism. The Arduino can easily be replaced by a higher speed micro controller, and varactors can replace stepper-motor controlled variable capacitors. The
ArduiTaM is made available in open source, so it is easily duplicated.

## 1 Introduction

Nuclear magnetic resonance spectrometers for imaging and spectroscopy are predominantly operated with inductor-capacitor (LC) resonators that pick up the radiofrequency signal due to the evolving spin magnetisation in a sample or patient. Because
the detected NMR signals are naturally very weak, it is of utmost importance that high quality-factor components are used along the analogue radiofrequency signal path, so as to minimize signal losses.

Proper tuning and matching (T&M) of the NMR probe is also important, and when done properly as will be demonstrated in the following section, this significantly enhances the receiver's sensitivity, thereby reducing the signal averaging times that would otherwise be required to obtain sufficient signal-to-noise ratio.
In many NMR/MRI systems, (T&M) is done manually by mechanically adjusting two trimmer capacitors. Although this hands-on process has been acceptable for many years, the need for automatic procedures has become important with the trend

---

[*]Shared first authorship.



towards high-throughput experiments, for example so that samples can be automatically loaded into the magnet and analysed without delay.

The major commercial system providers recently started offering add-ons to equip their probes with automatic tuning and

matching. These systems contain actuation units that drive the mechanical trimmer capacitors of the probe and are, in turn, controlled by the spectrometer software, which uses a so-called wobble routine to obtain the feedback signal. The offered solutions are costly and therefore only practical for probes with a limited number of channels. A further drawback is that the commercail systems are designed to drive mechanical trimmers, and therefore not adaptable to tuning and matching customized probes that for example involve digital capacitors or varactors. Moreover, these commercial systems are further limited by the

capabilities of the RF channels of the spectrometer, and thus lack the generality needed by the experimentalist. So for instance, an automatic (T&M) system installed in an NMR spectrometer with a $^1$H narrow-band channel cannot be used to (T&M) a $^{13}$C coil, or *vice versa*.

To overcome these limitations, we considered the availability of off-the-shelf analogue and digital electronic components needed for such a system. In this paper, we report our findings, and present a compact low-cost accurate Arduino-based auto-

matic tuning and matching system, abbreviated as ArduiTaM. Unlike commercial solutions and previously published reports Hwang (1998); Koczor (2015); Muftuler (2002); PerezdeAlejo (2004); Sohn (2015, 2013) that use either the MR spectrometer or a commercial network analyzer for frequency generation and signal processing, our system employs a single chip frequency source covering a useful range of $35\,\mathrm{MHz}$ to $4.4\,\mathrm{GHz}$, as well as discrete signal processing electronics, rendering it a completely standalone system capable of tuning and matching almost any relevant NMR/MRI probe channel. The ArduiTaM is

compatible with most NMR spectrometers, can readily be interfaced via a few TTL lines, and requires neither software add-ons (like Koczor's work Koczor (2015)) nor hardware alterations. The spectrometer-probe can thus be brought back to its original state by simply detaching the unit.

The ArduiTaM accomplishes (T&M) using the same principle that one would apply manually, namely, it monitors the signal reflected from the probe ($S_{11}$), then varies the tune and match capacitors until the minimum signal reflection amplitude

occurs at the Larmor frequency. This gives it the advantage of being completely independent of the probe's topology, thereby eliminating constraints on the probe, such as the orthogonality of tuning and matching considered in Hwang's paper Hwang (1998). Furthermore, the ArduiTaM can, apart from NMR/MRI experiments, be a handy tool in any RF laboratory, useful to characterize coils, coil arrays, antennas, and impedance matching networks.

## 2   Why tune and match?

In almost all commercial NMR systems the spectrometer electronics racks are placed adjacent to the magnet due to their relatively large dimensions. This necessitates the use of shielded coaxial cables to guide the NMR signals from the coil to the spectrometer. The coaxial cables usually have a characteristic impedance, $Z_0$, of $50\,\Omega$, and therefore all RF components of the NMR spectrometer are designed accordingly to ensure impedance matching and consequently maximise power transfer. The NMR coil is no exception, and the use of a $50\,\Omega$ coaxial cable to connect the coil to the spectrometer implicitly implies that





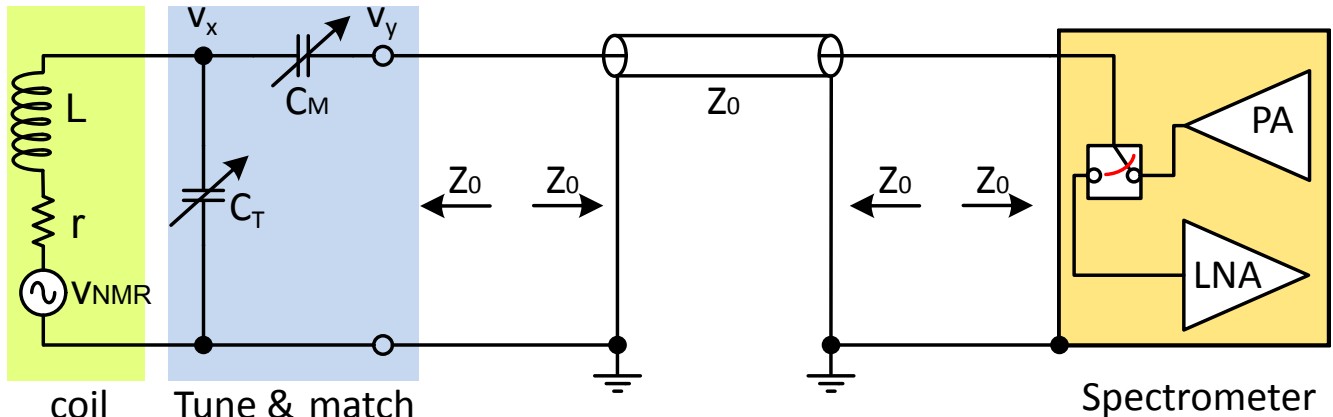

**Figure 1.** Example of a widely used parallel-tune/series-match topology. The L-matching circuit converts the complex impedance of the coil to a purely resistive impedance equal to the characteristic impedance of the coaxial cable, $Z_0$.

it also must be matched to $Z_0$ to guarantee an efficient excitation field, $B_1$. The impedance matching of the NMR coil also minimizes the signal reflection coefficient $\Gamma$

$$\Gamma = \frac{Z_{\text{coil}} - Z_0}{Z_{\text{coil}} + Z_0}, \tag{1}$$

and thus prevents the formation of standing waves in the cables Mispelter (2006).

Matching the coil impedance is commonly realized using high quality, almost lossless, capacitors, as illustrated in figure 1.

As such, the matching process also implies that the coil will be part of a resonant circuit, hence the term "tuning". The process of tuning and matching the coil in figure 1 can be performed as follows: first, $C_T$ is varied to make the coil resonate at a frequency, $\omega_r$, slightly above the Larmor frequency, $\omega_l$, such that the real impedance of the parallel resonator is $50\,\Omega$ at $\omega_l$. Then, the residual imaginary impedance, which in this case is inductive, can be eliminated by varying $C_M$. In effect, the extra capacitance is the complex conjugate of the residual inductance. This process results in the coaxial cable experiencing a purely

resistive $50\,\Omega$ load impedance at $\omega_l$. Due to resonance, the tuning and matching network acts, in what can be considered as one of its major advantages, as a passive noiseless preamplifier, thus $|v_y| = G_{\text{TM}} \cdot |v_{\text{NMR}}|$, with the tune and match voltage gain $G_{\text{TM}} = 0.5\sqrt{Z_0/r}$. According to Friis' formula,

$$F_{\text{Total}} = 1 + \frac{F_{\text{receiver}} - 1}{G_{\text{TM}}}, \tag{2}$$

this particular feature can significantly lower the noise factor $F$ (noise figure $\text{NF} = 10\log F$) of the NMR receiver and thus

enhance its sensitivity.

To obtain an impression of how significant this can be, consider the circuit in figure 2a as an example of a typical MR receive channel. The circuit uses the LMH6629 low noise amplifier from Texas Instruments® in a non-inverting topology. The operation frequency of the circuit is arbitrarily set to $500\,\text{MHz}$ and the T&M capacitors are assumed to be noiseless. The circuit was simulated using the Advanced Design System software (ADS) to explore the effect of T&M the NMR coil on the overall



signal-to-noise ratio. Figure 2b shows the voltage amplification, $G_{TM} = |v_y/v_{NMR}|$, due to T&M, while figure 2c demonstrates the ratio of the output SNR with T&M, to that without T&M, for different values of the coil's AC resistance. The later figure clearly shows how T&M can significantly enhance the SNR. The enhancement falls off as the coil's noise increases. For a coil with a relatively high AC resistance of, say $5\,\Omega$, T&M still results in an SNR enhancement by a factor of more than 2. Thus improper T&M results not only in an inefficient power transfer during transmission, but also in an increase in the NF during
reception, leading to a severe loss of sensitivity.

## 3 ArduiTaM implementation

The ArduiTaM requires hardware and software co-design, which is detailed in this section.

### 3.1 Circuit design

Figure 3 depicts the block diagram of the ArduiTaM circuit, for the case of T&M capacitors that are rotated using stepper
motors. This figure also describes how the ArduiTaM is easily inserted in the signal path of a spectrometer, underlining its integrability. It requires minimal connections, namely, a TTL trigger signal from the spectrometer to the Arduino, and a TTL acknowledgement signal from the Arduino back to the spectrometer, to facilitate automation. After loading a new sample, the spectrometer can thus trigger the ArduiTaM to readjust the T&M of the probe. Once the ArduiTaM is done, it will send an acknowledgment signal back to the spectrometer so as to start the NMR experiment. The ArduiTaM circuit consists of an
Arduino Uno as the master controller, carrying out the frequency sweep control, signal acquisition, signal processing, optimum T&M condition search, and the adjustment of the variable capacitors. The Arduino furthermore allows the user to set the channel Larmor frequency, as well as the frequency sweep range for tuning and matching. The ArduiTaM uses an ADF4351 high-quality ultra-wideband frequency synthesizer from Analog Devices®  to generate the required frequency sweeps. The synthesizer is controlled using an SPI protocol, and covers a frequency range from $35\,\mathrm{MHz}$ to $4.4\,\mathrm{GHz}$, making ArduiTaM
compatible with the frequencies of almost any commercial NMR spectrometer available today.

The output of the synthesizer is transferred to the NMR probe through a low-loss directional coupler (ZFDC-10-1 from Mini-Circuits®), and then through a high quality (low insertion loss and high isolation) RF switch (ZFSWA-2-46 from Mini-Circuits®). The latter is used to toggle between the "T&M mode" in which the NMR probe is connected to the ArduiTaM, and the "NMR mode" in which the probe is connected to the NMR spectrometer, and is the only additional lossy element that is
active during NMR signal acquisition (also refer to Figure 9).

The reflected signal from the probe is taken from the coupler and amplified using an ADL5611 from Analog Devices®  (all circuit schematics are available in the supplementary files). After that, the envelope of the reflected signal is extracted using a high sensitivity ($-40\,\mathrm{dBm}$) pin-diode-based envelope detector. The extracted envelope represents the probe's signal reflection, or $S_{11}$-curve, and is further amplified using an LM324 with an explicit low-pass filter of $200\,\mathrm{Hz}$ bandwidth to eliminate
high frequency residuals. The low frequency $S_{11}$-curve is acquired by the Arduino through one of its analog inputs, and the auto T&M algorithm is executed based on the acquired data. Figure 4 shows the interconnected discrete components of the

**MAGNETIC RESONANCE**
Open Access Discussions



**Figure 2.** The effect of T&M the NMR coil on the overall signal-to-noise ratio after the receiver. (a) Typical NMR receiver circuit including the coil, T&M capacitors, coaxial cable, and low-noise amplifier. The cable length was set to be short enough so that its attenuation is negligible. (b) The passive amplification, $G_{TM}$, as a function of the coil's resistance. (c) The overall SNR enhancement as a function of the coil's resistance. All simulations were performed using the Advanced Design System (ADS) software.

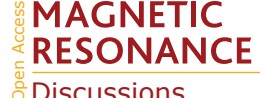

**Figure 3.** Block diagram of the ArduiTaM, detailing its component interconnections, as well as how the system can be inserted in the signal path of a spectrometer.

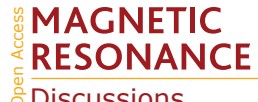



**Figure 4.** The experimental setup of the ArduiTaM showing the acquired $S_{11}$ signal of the coil on the oscilloscope.



**Figure 5.** (a) Four consecutive acquisitions of $S_{11}$, by ArduiTaM, of a coil with fixed $C_T$ value. The overlayed curves were acquired via one of the analog inputs of the Arduino. The x-axis covers the frequency range of the sweep. The y-axis represents the magnitude of the reflected signal. The insert zooms in onto the minimum of the curves. (b) $S_{11}$ of the same coil with four arbitrarily different $C_T$ values.

ArduiTaM in which an oscilloscope was used to visualize the acquired $S_{11}$ curve of the coil, verifying performance. Figure 5 displays four acquisitions of the $S_{11}$ curves recorded by the Arduino for two cases; when the tuning capacitor, $C_T$, of the coil is (a) fixed, and (b) arbitrarily changed. The results demonstrate the robustness and reliability of the circuit.




## 3.2 Tuning algorithm

Before and after the T&M protocol runs, a trigger signal is sent to and from the Arduino microcontroller to toggle between two modes of operation: T&M, and NMR measurement. Once the start trigger signal is received, the algorithm initializes all variables and resets the position of the capacitors, $C_t$ and $C_m$, to their minimum value using a so-called "homing routine". The routine causes the stepper motors to turn counterclockwise until they reach a lower angle limit, corresponding to the lowest value of the trimmer capacitors. Afterwards, the user is asked to enter the desired Larmor frequency, $f_0$, for which the tuning and matching conditions will be sought.

The algorithm sweeps the tuning capacitances against fixed values for the matching capacitances, capturing the $S_{11}$ values for each combination. $S_{11}$ values are read from an analogue input of the microcontroller at which the already amplified and extracted envelope signal is input. To reduce the algorithm's running time, the first sweep is done at a low resolution, with a sweep step size of $S_{sw} = 64 \cdot S_m$, where $S_m$ represents a single step of the stepper motor, which is equivalent to the inverse of the steps per turn, referred here as *spt*, that a stepper motor can provide. Our prototype uses motors with *spt* = 20, resulting in $S_m = 18°$. Once the first sweep is complete, the capacitor pair {$C_t$,$C_m$} with the lowest $S_{11}$ value ($S_{11min}$) is found and a new searching window is defined. The chosen seach intervals are symmetric around $S_{11min}$ with a span of $S_{sw}$ in each direction i.e., $S_{11min} \pm S_{sw}$, for which the corresponding tuning and matching capacitor values are already known. The same cycle is run four consecutive times, increasing the resolution by 4 with respect to the previous cycle until the maximum resolution of rotational increment is reached, that is, $S_{sw} = S_{sw}/4$ until $S_{sw} = S_m$ (see Fig. 6). Ultimately, the algorithm's resolution is limited by $S_m$, which can be easily improved with the use of stepper motors with a higher *spt* value, or through the use of a step down gear box. After the last cycle, the motors are positioned at the angles for which tuning and matching capacitances corresponded to the lowest probe reflection. Finally, the microcontroller activates the RF switch and sends a trigger signal to the NMR spectrometer to switch to NMR measurement mode.

## 4 Results

The ArduiTaM was first tested on a homemade RF coil. The solenoid was designed to operate in a $1.055\,\text{T}$ preclinical MRI magnet at a Larmor frequency of $44.93\,\text{MHz}$. We used two sapphire dielectric trimmers, each with 28-turn threads ($56\pi\,\text{rad}$), covering a range of $0.5\,\text{pF}$ to $10\,\text{pF}$, to tune and match the coil according to the circuit topology depicted in figure 2. Two stepper motors were used to mechanically drive the trimmers, see figure 7a. In this setup we explored two scenarios:

1. The two trimmers were arbitrarily set, and ArduiTaM was applied. Covering a tuning and matching range of 6 turns for each trimmer, and using 4 loops of resolution (corresponding to a final step of approximately $32\,\text{fF}$), ArduiTaM could precisely T&M the coil in $2\,\text{min}$ and $24\,\text{s}$. Figure 7b shows the $S_{11}$-curve of the coil before and after applying ArduiTaM.

2. The matching capacitor of the coil was adjusted to achieve an optimum matching at the Larmor frequency, whereas the tuning capacitor was arbitrarily set. Covering a wider sweep range of 9 turns with a finer resolution through 6 loops, ArduiTaM was capable of precisely tuning the coil, figure 7c, in $22\,\text{s}$.







**Figure 6.** (a) T&M flowchart. (b) Schematic showing the interval subdivision method to zoom in on the $S_{aa}$ minimum.

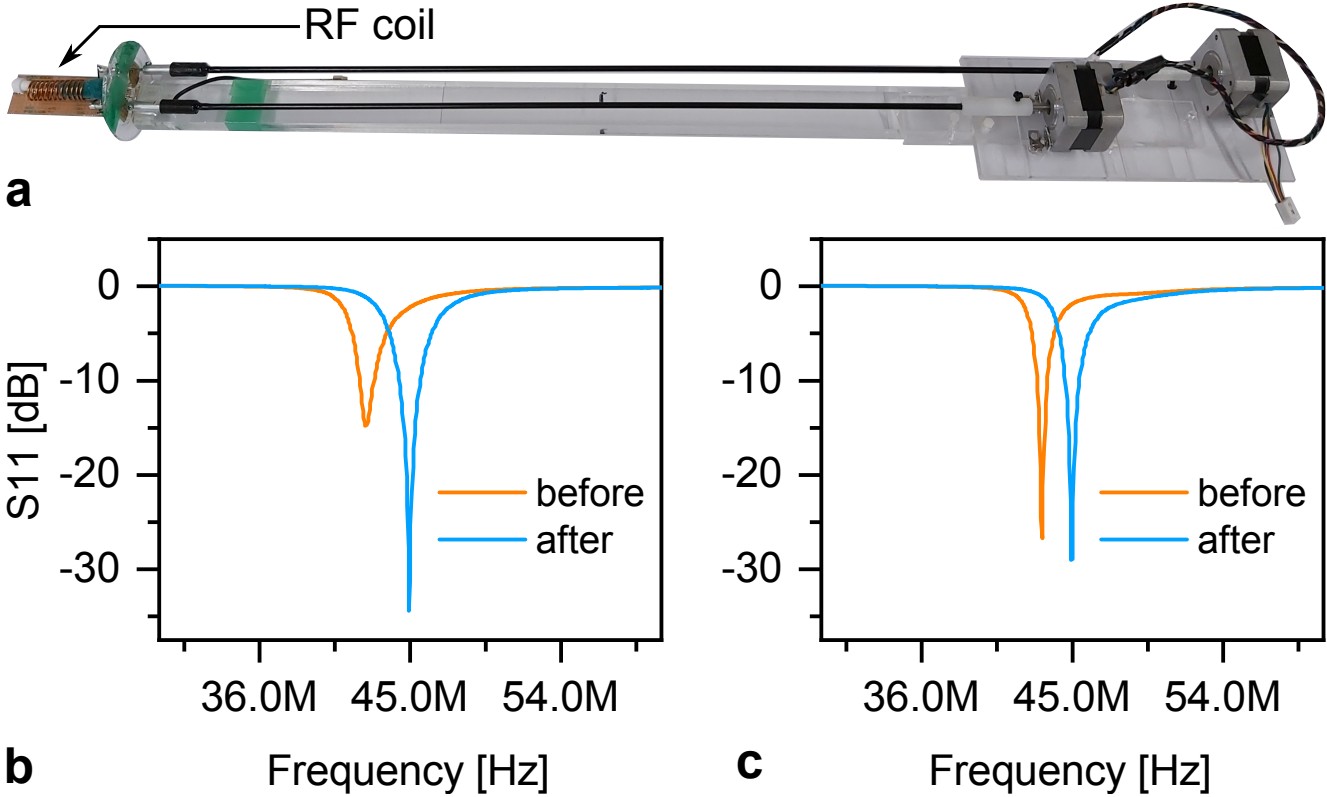

**Figure 7.** (a) Homemade RF coil with a mechanical setup for automatic T&M. The coil operates at a Larmor frequency of $44.93\,\mathrm{MHz}$. (b) Result of applying ArduiTaM to T&M the coil starting from an arbitrary condition. (c) Result of applying ArduiTaM to adjust the tuning capacitor only.

The speed of rotation of the stepper motors was kept conservatively low to protect the capacitors from mechanical damage, which affected the time of the measurement. The $S_{11}$ measurements in figure 7 were obtained using a Keysight E5071C network analyzer. In order to highlight its practicality, we applied ArduiTaM to a commercial probe (Bruker ICON) as shown
in figure 8a. The probe is designed such that its matching condition is always met over the entire range of targeted samples. Therefore, it exhibits a tuning capacitor only. We used an RF switch (ZFSWA-2-46), controlled by the TTL output of the MRI scanner, to toggle between the T&M mode where ArduiTaM runs, and the MRI mode in which the scanner's electronics are routed to the coil. Figure 8b plots the percentage of reflection from the coil, recorded using the standard "wobble" routine of the scanner, for three cases; first, when the coil was manually tuned, second, when the coil was arbitrarily de-tuned, and third,
when the coil was automatically re-tuned using ArduiTaM. The tuning range, the number of resolution loops, and thus the tuning time ($22\,\mathrm{s}$) were similar to the values used in figure 7c. In order to assess the effect of ArduiTaM on the performance of a commercial system, we conducted an imaging experiment of a cherry tomato with and without applying ArduiTaM, see figure 9. For both experiments, we utilized a standard gradient-echo sequence with a $1\,\mathrm{mm}$ slice thickness, $312\,\mathrm{\mu m}$ in-plane

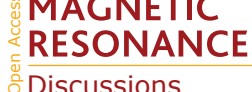

**Figure 8.** (a) Integrating ArduiTaM with a commercial MRI probe (Bruker ICON). (b) "Wobble" curves showing the coil's reflection when it is manually tuned, and before and after the automatic tuning using ArduiTaM.





**Figure 9.** Integrating ArduiTaM with a commercial MRI scanner shows only a negligible effect on its performance. (Left) MR image of a cherry tomato before applying ArduiTaM. The measured SNR is 49. (Right) Repeating the same experiment after applying ArduiTaM. The measured SNR here is 43, reflecting a loss of approximately $12\,\%$, due to the insertion loss of the RF switch (ZFSWA-2-46 from Mini-Circuits®).

resolution, $30°$ flip angle (FA), $4\,\text{ms}$ echo time (TE), $100\,\text{ms}$ repetition time (TR), and averaging 8 times. Moreover, to ensure

that both experiments have exactly the same initial settings, we ran the automatic adjustment routines before each experiment. These include a "drift adjustment" to correct the frequency drift caused by the field drift, a "power adjustment" to calculate the correct power that corresponds to the desired flip angle, and a "receiver gain adjustment" to calculate the receiver gain that ensures optimum digitization of the signals. The measured SNR before applying ArduiTaM, figure 9(left) is 49, while the SNR after inserting ArduiTaM, figure 9(right), is 43 corresponding to a $12\,\%$ loss in the signal-to-noise ratio. The slight loss in SNR

is due to the insertion loss of the RF switch ($0.8\,\text{dB}$) needed by ArduiTaM to route the RF signals.

## 5 Conclusions

ArduiTaM is a fully functional automatic tuning and matching system for use in NMR or MRI systems. Despite its simplicity, ArduiTaM exhibits high precision, reliability, and most importantly, a negligible influence on NMR signal fidelity. By utilizing an ADF4351 frequency synthesizer, ArduiTaM is completely independent of the spectrometer, and conveniently covers a

**MAGNETIC RESONANCE** Discussions

frequency range of $35\,\mathrm{MHz}$ to $4.4\,\mathrm{GHz}$, making it compatible with most commercial NMR or MRI frequencies. Moreover, its very low cost (around 100 Euros for the chip components at the time of writing) makes it particularly attractive for use with multi-resonant probes and MRI phased arrays. Away from MR, other potential applications, such as antennas, resonators, and impedance matching networks, may stand to benefit as well.

ArduiTaM was successfully tested on a homemade coil prototype that uses stepper-motor-driven trimmer capacitors for
tuning and matching. Trimmers were purposely employed to demonstrate ArduiTaM's ease of integrability with commercial systems, where such capacitors are predominantly used. Unfortunately, the use of mechanical trimmers imposes a limit on the speed of T&M. However, trimmers can easily be replaced with varactors or digitally-tuned capacitors (DTCs), for much faster tuning and matching. The speed of T&M can be further optimized depending on the experiment. So, for example, if the maximum frequency shift happens to be known for the samples under test, then the sweep ranges can be minimized so that
ArduiTaM can make more rapid adjustments.

*Code and data availability.* All circuit schematics are provided in the supplementary file. The Arduino codes are available under: https://github.com/mazinjouda/ArduiTaM.git/.

*Author contributions.* Concept initialisation: MJ and JGK. Circuit design, implementation, and test: MJ. Tune and match algorithm: SMTD and DM. Writing the Arduino codes: SMTD. Building and testing the mechanical setups: MAJ. Final measurements: MJ. Funding request
and supervision: JGK. Writing, review, and editing: MJ, SMTD, MAJ, DM, and JGK.

*Competing interests.* The authors declare no competing interests.

*Acknowledgements.* JGK and MJ acknowledge the DFG for partial funding [contracts KO 1883/29-1 and KO 1883/34-1]. ST, MJ, DM, and JGK would like to thank the European Union for partial funding [Grant H2020-FETOPEN-1-2016-2017-737043-TISuMR].





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
