# Peer review of "ArduiTaM: accurate and inexpensive NMR auto tune and match system"

_Magnetic Resonance, 2020_

## Referee Comment (RC1) · Norbert Mueller (Referee) · 3 May 2020

General remarks: This contribution offers a way of automated tuning and matching in NMR and MRI using an open-source hardware and software solution with a single board micro-controller. This maybe a good approach for from scratch new system design and refurbishing of older systems. With current commercial NMR probes we are not sure such a system can help to work around some of the commonly encountered problem, that is mechanical hysteresis and slip of the controller rods connecting the stepper motors to the capacitors. A software that "learns" this non ideal behavior of probes (which is also subject to aging) and can manipulate the actuators more efficiently than a human would be nice. In a production environment (i.e. with high throughput). The method for tuning and matching is not fundamentally "new" as it

relies on the well known rf-responses exploited by current commercial hardware. The speed of the process is apparently better, but it is not clear, if the same goal couldn't be achieved by an adaptive software implementation with existing hardware. Where the proposed solution may be highly beneficial are probes going beyond the current mechanical trimmer capacitor based system using varactors or digital capacitors, as the authors emphasize. NB: I have been looking at the paper mostly from a spectroscopy perspective. Some of my arguments may not apply in an imaging context.

Specific questions and comments: In the "homing routine" for driving the capacitors to their lowest values: How is the stop detected? Steppers do not usually sense that they are stuck and absolute positioning may not maintained when restarting the Arduino. Is there absolute angle sensing on the steppers? Is there a provision against excessive torque being applied? About the steppers: 20 steps per turn sounds like a low number. Could a gearbox be added achieve lower angles. But that could increase the mechanical instability. The first scan of the two capacitors appears to be quite coarse. Depending on the type of probe (high resolution liquids vs. static solids, for example you may to miss the actual minimum (multiply tuned probes may have several minima). How does the algorithm handle the sudden "jumps" when changing the capacitances (as experienced in manual tuning T&M, likely caused by release of torsional stress in long connection rods? This is a big problem, also for commercial systems and is (in my opinion) the reason for suboptimal T&M in existing systems. Could the hardware/software combo be extended to other tuning/matching modalities, like frequency pushing effects or spin noise? [J Mag Res 193 (2008) 153; J Biomol NMR 45 (2009) 241; ChemPhysChem 15 (2014) 3639] Considering the fact that the impedance of the pre-amplifier and the transmission line from the rf-coil to the pre-amp play a major role in high efficiency tuning, the scope of the approach, which apparently requires switching between the envelope detector and the proper detection preamplifier may be applicable to, for example, cryogenically cooled probes.

Conclusion: The paper offers new ideas to approach the practical problems of automatic tuning, but some additional discussion addressing the issues mentioned above would increase its impact. It is probably more of a technology demonstration than purely scientific innovation. In my opinion, the paper will be of largest benefit for researchers designing new probes in the field of imaging and solid state NMR. It may also be a starting point for designing future high resolution / high sensitivity NMR probe systems.

―――――――――――――――――――――――――

---

## Referee Comment (RC2) · Anonymous Referee #2 · 14 May 2020

The authors claim an automatic system for RF tuning and matching through a micro-controller (Arduino) and variable capacitors by mechanical step-motors. This reviewer acknowledges the need to develop automatic system to replace time-consuming manual adjustment. However, some points are not clear and mislead the readers so a minor revision is necessary to improve the quality of this manuscript to be published. Major and minor concerns are listed below.

Major concerns 1. This study is limited in receive coil only. As the authors state in part 2 (why tune and match?), the tuned and matched condition is matter even at receiving side. However, the effect of impedance matching at the transmit side is more important in SNR. There is no any comments about the transmit side, tuning and matching condition with sample. How can the authors verify the condition of the transmit is always

[Figure]

same during all experiments for this work. Perhaps, dose it relate to #5 in minor concerns? Even very small difference at transmit side (distance between a sample and coil or between transmit and receive coil. . ...) make a huge different results in S-parameters (S11 and S21) that ultimately make different SNR. The authors should explain how to keep the same condition to conduct the comparison of receive only coil among manual, before and after auto tune and match. 2. Tuning and matching, in general, are performed after a sample is loaded. If the shift by loading a sample s negligible, this work has less scientific impact. Also, RF coils that have a little bit wide bandwidth may resolve the loading problems. In this respect, the authors are required supplement addition experiments to prove the necessity of this work. For example, bench test results with a universal tuning and matching before and after a sample loads to show loading effect that should be compensated. Also, the authors can add an image and SNR comparison with a universal tuned and matched coil using a sample without manual and auto correction in figure 9. 3. (Line 7, page2) and (Line 94, page 4): it misleads the readers because VCO may cover the frequency range for 1T to 23T but it is almost impossible to adjust resonating frequency (i.e., Larmor frequency) with capacitors and RF coil element. Otherwise, the authors should provide additional description with practical values (e.g., capacitor values and/or size of resonating structure) that can over the wide range of NMR system.

Minor concerns 1. Is stepper motor okay for MRI study even if it has strong magnetic property? It may bring safety issues or magnetic field disturbance. 2. (Line 24, page 2): Please add a reference in the bibliography about the commercial system. 3. How can the NMR console receive a signal from ArduiTaM system? Is there an open port in the console or the homebuilt system need an interface module? The details should be described in the manuscript to better understanding because the minimized interface between automatic system and NMR console is important in this application. 4. There are some typo (e.g., line 28, page2) and improper English grammar/expression. The authors need to carefully check it. 5. This reviewer would like to ask to the authors to add the mathematical analysis to explain SNR comparison in Line 158, page 13.

How only 0.8 dB insertion loss results in lowering 6 dB in SNR in figure 9. 6. Figure 5 missed the unit for the Y-axis.

---

## Author Comment (AC1) · 14 May 2020

Authors: First of all, we would like to thank the reviewer for giving the time and effort to reviewing our manuscript.

Reviewer: General remarks: This contribution offers a way of automated tuning and matching in NMR and MRI using an open-source hardware and software solution with a single board microcontroller. This maybe a good approach for from scratch new system design and refurbishing of older systems. With current commercial NMR probes we are not sure such a system can help to work around some of the commonly encountered problem, that is mechanical hysteresis and slip of the controller rods connecting the stepper motors to the capacitors. A software that "learns" this non ideal behav-

ior of probes (which is also subject to aging) and can manipulate the actuators more efficiently than a human would be nice. In a production environment (i.e. with high throughput).

Authors: This is a very interesting idea, although the ideal solution would be to make the mechanical system (trimmers, rods, etc.) robust against aging and hysteresis, if not permanently then at least for some long enough time after which the mechanical system can be renewed. Nevertheless, the idea of involving "learning" algorithms is very motivating and can definitely provide solutions to circumvent the mechanical hysteresis problem, and can probably make T&M faster. Just like the difference between a new NMR user and an experienced user.

Reviewer: The method for tuning and matching is not fundamentally "new" as it relies on the well known rf-responses exploited by current commercial hardware. The speed of the process is apparently better, but it is not clear, if the same goal couldn't be achieved by an adaptive software implementation with existing hardware. Where the proposed solution may be highly beneficial are probes going beyond the current mechanical trimmer capacitor based system using varactors or digital capacitors, as the authors emphasize. NB: I have been looking at the paper mostly from a spectroscopy perspective. Some of my arguments may not apply in an imaging context.

Reviewer: Specific questions and comments: In the "homing routine" for driving the capacitors to their lowest values: How is the stop detected? Steppers do not usually sense that they are stuck and absolute positioning may not maintained when restarting the Arduino. Is there absolute angle sensing on the steppers? Is there a provision against excessive torque being applied?

Authors: The current implementation requires the user to do one-time single-point calibration of the tuning and matching capacitors when attaching ArduiTaM for the first time. This can be easily done by manually rotating the trimmers to their lower value which is recognized by ArduiTaM as the zero position, and by setting the allowed number of turns before the upper value is reached. As long as it is not turned off, ArduiTaM can keep track of stepper motors' positions and ensure that neither limit is exceeded. A simple modification of the code, such that current positions the trimmers are written in the EEPROM of the Arduino, would allow ArduiTaM to keep track of the stepper motors' positions even after being completely powered off. If this one-time single-point calibration step is to be avoided, then a couple of limit switches for the tuning and matching trimmers would be required.

Reviewer: About the steppers: 20 steps per turn sounds like a low number. Could a gearbox be added achieve lower angles. But that could increase the mechanical instability. The first scan of the two capacitors appears to be quite coarse. Depending on the type of probe (high resolution liquids vs. static solids, for example you may to miss the actual minimum (multiply tuned probes may have several minima).

Authors: We are extremely sorry for this typo. In fact the stepper motors we used are SM-42BYG011-25 and have a 1.8 degree/step resolution, thus requiring 200 steps per turn. With this high resolution, it is very unlikely to miss the actual minimum, and that is probably why we never encountered this problem during the experiments. Nevertheless, this is a very valid point and suggests that one has to be very careful when selecting a stepper motor or defining the sweep resolution. Indeed, having a higher stepping resolution and a finer sweep would likely guarantee finding the minimum, but that is unfortunately at the expense of prolonged T&M time.

Reviewer: How does the algorithm handle the sudden "jumps" when changing the capacitances (as experienced in manual tuning T&M, likely caused by release of torsional stress in long connection rods? This is a big problem, also for commercial systems and is (in my opinion) the reason for suboptimal T&M in existing systems.

Authors: In fact we did not encounter this problem during the experiments. This is probably because the mechanical setup was newly built and we used new high quality trimmers and stiff glass fiber rods. Another potential reason is the relatively low operat-

ing frequency (45 MHz) where slight capacitance changes are less pronounced. From our experience, we know that trimmer capacitors become loose after going through many tuning cycles. This is most likely the reason behind these jumps. In such a case, the ideal remedy would be to replace the trimmers with fresh ones. Alternatively, one can think of a software solution by always comparing three or four consecutive S11 points and check for dramatic jumps so that incorrect points can be reacquired.

Reviewer: Could the hardware/software combo be extended to other tuning/matching modalities, like frequency pushing effects or spin noise? [J Mag Res 193 (2008) 153; J Biomol NMR 45 (2009) 241; ChemPhysChem 15 (2014) 3639] Considering the fact that the impedance of the pre-amplifier and the transmission line from the rf-coil to the pre-amp play a major role in high efficiency tuning, the scope of the approach, which apparently requires switching between the envelope detector and the proper detection preamplifier may be applicable to, for example, cryogenically cooled probes.

Authors: The short answer is yes! The long answer is as follows; ideally, both the output impedance of the power amplifier and the input impedance of the preamplifier should be exactly equal to the characteristic impedance of the coaxial cables (usually 50 Ohm). If this is the case, then matching the coil in the excitation phase by minimizing its reflected power necessarily means that the coil will also be matched to the preamplifier in the reception phase. Practically, these impedances as well as the characteristic impedance of the coaxial cable might slightly differ from 50 Ohm resulting in slight discrepancy between the matching conditions in the excitation and reception phases. Usually this discrepancy is small and thus negligible. However, it might become significant in certain circumstances as for example high frequency systems and high Q probes. In such cases, one solution to tackle this problem is to measure this discrepancy on the spectrometer where ArduiTaM is to be installed and to feed this measurement as an offset frequency to the ArduiTaM such that when the minimum reflection is achieved in the excitation phase at the offset frequency the matching condition is fulfilled in the reception phase at the Larmor frequency. This should work

robustly as long as the Larmor frequency does not drift considerably.

Conclusion: The paper offers new ideas to approach the practical problems of automatic tuning, but some additional discussion addressing the issues mentioned above would increase its impact. It is probably more of a technology demonstration than purely scientific innovation. In my opinion, the paper will be of largest benefit for researchers designing new probes in the field of imaging and solid state NMR. It may also be a starting point for designing future high resolution / high sensitivity NMR probe systems.
* * *

---

## Author Comment (AC2) · 23 May 2020

Authors: First of all, we would like to thank the reviewer very much for investing time and effort to review our manuscript.

Reviewer: General comments: The authors claim an automatic system for RF tuning and matching through a micro- controller (Arduino) and variable capacitors by mechanical step-motors. This reviewer acknowledges the need to develop automatic system to replace time-consuming man- ual adjustment. However, some points are not clear and mislead the readers so a minor revision is necessary to improve the quality of this manuscript to be published. Major and minor concerns are listed below.

Reviewer: Major concerns 1. This study is limited in receive coil only. As the authors

state in part 2 (why tune and match?), the tuned and matched condition is matter even at receiving side. However, the effect of impedance matching at the transmit side is more important in SNR. There is no any comments about the transmit side, tuning and matching condition with sample.

Authors: In fact, the study is general and was applied to transceive (transmit/receive) coils. We are sorry if the relevant paragraph was not completely clear to convey that message. But in lines 50-55 we mention that the impedance of the coil and the spectrometer electronics (this implicitly includes transmitter and receiver) should match the characteristic impedance of the coaxial cables. Moreover, in that paragraph we stated that matching the coil to 50 Ohm maximizes the power transfer and thus guarantees efficient excitation field B1. We definitely meant here the excitation phase of the NMR experiment. Furthermore, the experiments reported in this paper were all performed on transceive coils.

Reviewer: How can the authors verify the condition of the transmit is always same during all experiments for this work. Perhaps, dose it relate to #5 in minor con- cerns? Even very small difference at transmit side (distance between a sample and coil or between transmit and receive coil. . ...) make a huge different results in S-parameters (S11 and S21) that ultimately make different SNR. The authors should explain how to keep the same condition to conduct the comparison of receive only coil among manual, before and after auto tune and match.

Authors: We think that this point was based on the assumption that we used different coils for Tx and Rx. If this were the case, then this concern would have been very valid and then careful attention must be paid to ensure that the transmitter conditions do not change. But as we mentioned in the first comment, in our experiments we used a single transceive coil. As such and as long as the sample does not change the transmit conditions will not change.

Reviewer: 2. Tuning and matching, in general, are performed after a sample is loaded.

If the shift by loading a sample s negligible, this work has less scientific impact. Also, RF coils that have a little bit wide bandwidth may resolve the loading problems. In this respect, the authors are required supplement ad- dition experiments to prove the necessity of this work. For example, bench test results with a universal tuning and matching before and after a sample loads to show loading effect that should be compensated. Also, the authors can add an image and SNR com- parison with a universal tuned and matched coil using a sample without manual and auto correction in figure 9.

Authors: As detailed in Hoult's paper (Hoult, D. I. (1979). Journal of Magnetic Resonance, 34, 425–433.) the sample-loading effect depends on many parameters including the coil geometry, coil parameters (R, L, C, Q), sample parameters, and the frequency of operation. To elaborate how large this effect can be, we did a couple of experiments and added two figures to the supplementary materials, which we also attach here as Fig 1 and Fig 2.

Figure 1: shows the unloaded/loaded S11 of a commercial 5 mm saddle coil designed to operate in a 11.7 T NMR magnet at 500 MHz. The sample we used was 0.5 M NaPO3, 0.5 M Phosphoric Buffer Solution (PBS), 50 mM TSP, and 0.5 MSucrose. Loading the sample showed a dramatic change (6 MHz frequency shift and 28 dB increase in the reflection) in the tuning and matching condition.

Figure 2: shows the NMR spectrum of the sample before and after readjusting the T&M condition. Readjusting the T&M condition showed a significant enhancement (6.4 times) of the SNR.

Reviewer: 3. (Line 7, page2) and (Line 94, page 4): it misleads the readers because VCO may cover the frequency range for 1T to 23T but it is almost im- possible to adjust resonating frequency (i.e., Larmor frequency) with capacitors and RF coil element. Otherwise, the authors should provide additional description with practi- cal values (e.g., capacitor values and/or size of resonating structure) that can over the wide range of NMR system.

Authors: Sorry for this confusion! What we wanted to say is that ArduiTaM can be used to automatically T&M almost every commercial probe. This is because it utilizes a programmable frequency synthesizer (ADF4351) capable of generating any frequency in the range from 35MHz to 4.4 GHz. So for example, ArduiTaM can T&M any of the following: a 300 MHz 1H probe in a 7T magnet, a 125 MHz 13C probe in a 11.7T magnet, a 600 MHz 1H probe in a 14.1T magnet, and so on. In all these examples, the T&M ranges are limited mainly by the trimmers of the probe (maximum tuining range is in the order of few MHz).

Reviewer: Minor concerns 1. Is stepper motor okay for MRI study even if it has strong magnetic property? It may bring safety issues or magnetic field disturbance.

Authors: If the magent is well shielded and if the motor is placed far enough then this shouldn't be a problem. In our experiment we used a 1T MRI permanent magnet and we placed the stepper motor outside the magnet where the average stray field is 500 uT. We didn't observe any forces or malfunction of the motor. Indeed, the Bruker sample changer system also uses similar stepper motors.

Reviewer: 2. (Line 24, page 2): Please add a reference in the bibliography about the commercial system.

Authors: Done.

Reviewer: 3. How can the NMR console receive a signal from ArduiTaM system? Is there an open port in the console or the homebuilt system need an interface module? The details should be described in the manuscript to better understanding because the minimized interface between automatic system and NMR console is important in this application.

Authors: From our experience of working with different NMR spectrometers, almost all systems provide a number of general purpose TTL inputs and outputs that can be easily programmed by the user to do various contol and triggering tasks. For example, the

Bruker preclinical MRI machine (ICON) has one TTL output trigger port and one TTL input trigger port. For imaging experiments using ParaVision, the TTL output trigger can be set in the pulse program using the commands "TTL1_HIGH" and "TTL1_LOW", while the TTL input can be used (by ticking a Checkbox in the GUI) to allow an external signal to trigger the signal acquisition.

Reviewer: 4. There are some typo (e.g., line 28, page2) and improper English grammar/expression. The authors need to carefully check it.

Authors: Checking done.

Reviewer: 5. This reviewer would like to ask to the authors to add the mathematical analysis to explain SNR comparison in Line 158, page 13.

Authors: The formula we used to calculate the SNR loss due to ArduiTaM is SNR_loss = (SNR(without ArduiTaM)-SNR(with ArduiTaM))/SNR(without ArduiTaM)*100

Reviewer: How only 0.8 dB insertion loss results in lowering 6 dB in SNR in figure 9.

Authors: The SNR values mentioned in this figure are in linear scale. Thus, the loss in SNR is around 1.13 dB.

Reviewer: 6. Figure 5 missed the unit for the Y-axis.

Authors: We have changed this figure.

———————————————————

[Figure]

**Fig. 1.** Sample loading effect

**Fig. 2.** Before and after effect